# Effectiveness of Integrative Korean Medicine Treatment in Patients with Traffic-Accident-Induced Acute Low Back Pain and Mild Adult Scoliosis

**DOI:** 10.3390/healthcare11121735

**Published:** 2023-06-13

**Authors:** Nayoung Shin, Hyejin Nam, Dong Woo Kim, Yoon Jae Lee, Doori Kim, In-Hyuk Ha

**Affiliations:** 1Department of Korean Medicine Rehabilitation, Ulsan Jaseng Hospital of Korean Medicine, 662-9, Sinjeong-dong, Nam-gu, Ulsan 44676, Republic of Korea; 2Department of Korean Medicine Ophthalmology, Otolaryngology & Dermatology, Ulsan Jaseng Hospital of Korean Medicine, 662-9, Sinjeong-dong, Nam-gu, Ulsan 44676, Republic of Korea; sppai@jaseng.org; 3Department of Internal Korean Medicine, Ulsan Jaseng Hospital of Korean Medicine, 662-9, Sinjeong-dong, Nam-gu, Ulsan 44676, Republic of Korea; chn19@jaseng.org; 4Jaseng Spine and Joint Research Institute, Jaseng Medical Foundation, Gangnam-gu, Seoul 06110, Republic of Korea; goodsmile@jaseng.org

**Keywords:** acute low back pain, traffic accident, scoliosis, Korean medicine, survey

## Abstract

We investigated the effectiveness of integrative Korean medicine treatment in patients with pre-existing scoliosis who received inpatient care for traffic-accident-induced acute LBP. We selected 674 patients diagnosed with scoliosis between 1 January 2015, and 30 June 2021, using lumbar spine (L-spine) imaging, across four Korean medicine hospitals in Korea for a retrospective chart review and sent them a questionnaire-based follow-up survey. The primary outcome was a numeric rating scale (NRS) score of LBP. The secondary outcomes were the Oswestry Disability Index (ODI), 5-level EuroQol 5-dimension (EQ-5D-5L), and patient global impression of change (PGIC) scores. In total, 101 patients responded to the follow-up survey. NRS scores decreased from 4.86 (4.71–5.02) to 3.53 (3.17–3.90) from admission to discharge, subsequently decreasing to 3.01 (2.64–3.38) (*p* < 0.001) at the last follow-up. Similarly, ODI scores decreased from 35.96 (33.08–38.85) to 22.73 (20.23–25.24) and 14.21 (11.74–16.67) (*p* < 0.001), respectively. Approximately 87.1% of patients were satisfied with their inpatient care. There were no significant differences in the degree of improvement according to the severity of scoliosis. Integrative Korean medicine treatment can improve pain, lumbar dysfunction, and quality of life in patients with traffic-accident-induced acute low back pain and pre-existing mild scoliosis.

## 1. Introduction

According to the Korea Road Traffic Authority [1], the number of registered motor vehicles in Korea increased from 20,989,885 in 2015 to 24,911,101 in 2021, indicating a recent increase in motor vehicle ownership per household. The number of traffic accidents (TAs) reached 203,130 in 2021, resulting in 2916 deaths and 291,608 injuries, with TA-induced injuries and management of sequelae emerging as important issues. Various musculoskeletal injuries, internal contusions, and psychological trauma can be observed after a TA [2,3], with many patients subsequently developing neck and low back pain (LBP) [4,5]. According to the Korean Statistical Information Service [6], patients with cervical sprain and strain accounted for the highest percentage of patients who received outpatient care after TAs in 2020 (n = 1,136,186), followed by those with lumbar/pelvic sprains and strains (n = 642,291). TA victims have a higher risk of LBP than non-TA victims [5]. Cassidy et al. [7] reported that recovery from LBP following TA is often delayed or incomplete, and 31% of patients still experience LBP after 12 months. Another cohort study [8] reported that lower back injuries caused by TAs might be a risk factor for chronic LBP, noting that TA victims with lower back injuries tended to experience recurrent LBP episodes.

Scoliosis, the most common spinal deformity affecting adults, is a three-dimensional deformity with loss of the sagittal curve accompanied by lateral curvature, coronal deviation of the spine, and vertebral rotation [9]. In 2021, the number of patients with scoliosis in Korea was approximately 94,000 [10], while a study of American adults aged 25–74 years reported a scoliosis prevalence of 8.3% between 1971 and 1975 [11], with a prevalence almost twice as high among women than men (10.7% vs. 5.6%). In another study [12], the prevalence of scoliosis among adults aged ≥40 years was 8.85%, and the authors noted that it increased with age; however, sex was not a significantly influential factor.

Adult scoliosis is divided into adult idiopathic scoliosis and adult degenerative scoliosis [13]. However, it is difficult to clearly distinguish between idiopathic scoliosis and degenerative scoliosis if the patient is not sure when the transformation began [14]. According to one study, the prevalence of primary degenerative scoliosis was approximately 38%, more common in women than men (41.2% vs. 27.5%), and with a higher prevalence in those over 60 years of age (36% vs. 13%) [15]. Idiopathic scoliosis is classified according to the patient’s age at the time of diagnosis, and can be subdivided into three types. Infantile scoliosis is divided into under 3 years old, juvenile scoliosis is 3–9 years old, and adolescent scoliosis is 10–18 years old [13]. Degenerative scoliosis is a scoliosis developed after adulthood, and the age standard is ambiguous. One study of degenerative scoliosis was based on a minimum age of 45 [16], and another study was based on a minimum age of 50 [17]. In a different way, scoliosis is divided into structural scoliosis and non-structural (antalgic) scoliosis. Structural scoliosis is a lateral curvature that does not correct completely on lateral bending. It has an asymmetry in the vertebrae, muscles, or ligaments of the spine itself and is almost always accompanied by a rotational deformity. Non-structural scoliosis can be completely corrected by lateral bending and does not involve fixed rotational deformity, and the cause is inconstant or asymmetry of the supporting skeleton [18]. Patients with scoliosis experience pain and neurological symptoms that affect their quality of life (QoL) and physical functioning, as well as various other symptoms, including progressive incurvation of the spine [19,20,21]. Among these symptoms, the most problematic for adults with scoliosis-related LBP is muscle fatigue and muscle spasm on the convex side and neuromuscular stimulation on the concave side [22,23]. A study [24] reported that patients with scoliosis tended to have more severe and longer-lasting LBP than those without scoliosis, with the pain level increasing with age and the extent of lateral incurvation.

One of the most common criteria for scoliosis is the Cobb angle [25], which was originally proposed by the American orthopedic surgeon John Robert Cobb. The Scoliosis Research Society (SRS), founded in 1966, has officially adopted the Cobb angle as the standard quantification of scoliosis [26]. The minimum angle for defining scoliosis is usually based on a Cobb angle of 10° [27,28]. Many studies have demonstrated the effectiveness of Korean medicine (KM) treatment for scoliosis and acute LBP. Wei et al. [24] reported a significant decrease in the Cobb angle after 12 months of therapy incorporating traditional Chinese medicine in patients with scoliosis. Moreover, Liu et al. [29] reported that 6 weeks of integrative treatment, including acupuncture, electropuncture, and moxibustion, reduced LBP pain in patients with scoliosis by >50%. Hasegawa et al. [30] further demonstrated that acupuncture therapy significantly reduced the visual analog scale (VAS) score for pain and the use of anti-inflammatory agents within the first 2 weeks of treatment in patients with acute LBP compared with results observed in the control group. In a similar study of patients with LBP, Kennedy et al. [31] reported significant reductions in the VAS score and the use of analgesics in the intervention compared with values observed in the control group after 3 months of acupuncture therapy.

Several studies have investigated the effectiveness of KM treatment for acute LBP and scoliosis. However, no studies have examined the efficacy of integrative KM treatment in patients with TA-induced acute LBP and pre-existing scoliosis. To address this, we aimed to compare therapeutic effectiveness, stability, patient satisfaction, and differences in improvement according to scoliosis severity among patients with TA-induced acute LBP and pre-existing scoliosis.

## 2. Materials and Methods

### 2.1. Study Design

In this study, we retrospectively analyzed electronic medical records (EMRs) and collected questionnaire responses for patients aged 19–65 years with radiological findings of scoliosis (Cobb angle ≥10°). Participants were selected from patients who received inpatient TA-induced LBP care across four Korean hospitals between 1 January 2015 and 30 June 2021. The four KM hospitals included in the study are accredited and designated by the Korean Ministry of Health and Welfare as specialists in spinal disorders. All hospitals provided KM treatment based on modern medical diagnostic technologies. Based on imaging data, a radiologist and a trained KM doctor (KMD) evaluated findings related to scoliosis and Cobb angle.

EMRs were reviewed for relevant parameters during the hospital stay, including personal information, type and duration of treatments, range of motion (ROM), and severity of scoliosis. Only data from the initial hospitalization were used for patients with multiple hospitalizations within the study period. Follow-up surveys were conducted through telephone and Google Forms between May and June 2022. The follow-up survey assessed the degree of improvement immediately after integrative KM treatment, current LBP pain and disability levels, recent treatments, scoliosis surgery, use of a brace, treatment satisfaction at the KM hospital, and preferences for Western medicine (WM) and KM treatments.

This study was approved by the Institutional Review Board (IRB) of the Jaseng Hospital of Korean Medicine (approval no.: JASENG 2022-04-008; approval date: 16 June 2022) and was conducted following the Declaration of Helsinki. The study protocol and manuscript were written per STROBE guidelines.

### 2.2. Participants

#### 2.2.1. Inclusion Criteria

The inclusion criteria were as follows: (1) 19–65 years of age, (2) receipt of inpatient care at the study hospitals for the chief complaint of TA-induced LBP, (3) radiological (lumbar spine) findings of scoliosis and Cobb angle ≥10°, and (4) provision of voluntary verbal consent to participate in the study. Cobb angle was measured as the largest angle formed by the superior endplate of the superior end vertebra and the inferior end vertebra’s inferior endplate on anteroposterior lumbar radiographs (Figure 1) [32].

#### 2.2.2. Exclusion Criteria

The exclusion criteria were as follows: (1) duration of hospitalization ≤3 days, (2) LBP with a diagnosed disease as the clear cause, (3) presence of other chronic diseases that may interfere with the interpretation of therapeutic effects or outcomes, (4) LBP due to a soft-tissue rather than spinal abnormality, and (5) presence of progressive neurological defects or severe neurological symptoms. Patients determined by the researchers to be ineligible to participate in the study and those who did not provide consent were also excluded.

### 2.3. Treatment

During the inpatient stay, patients received integrative KM treatment consisting of herbal medicine, acupuncture, pharmacopuncture, Chuna, moxibustion, and cupping therapy. Data related to these treatments were collected from EMRs.

Chuna therapy is a manual form of KM therapy in which a KMD uses the hand, a different part of the body, or an assistive device to stimulate the body’s physical structure. Chuna therapy was performed daily for approximately 5–10 min before acupuncture therapy during the hospital stay. The method was determined by trained KMDs who had completed the standardized Chuna education course.

For pharmacopuncture therapy, a standardized disposable insulin syringe (29 G × 13 mm, 1 mL, Sungshim Medical, Bucheon, Korea) was used to inject 1 mL of herbal medicine into the acupoints. The injection was administered once a day into selected acupuncture points and Ashi points according to the judgment of the KMD. By identifying the patterns of the symptoms of the patient, Shinbaro and Coptidis rhizome pharmacopuncture therapies were administered prior to the acupuncture.

Disposable needles (0.25 mm × 30 mm; DongBangAcupuncture Needles, Seongnam, South Korea) were used to perform acupuncture therapy by needling Ashi points and BL23 (Shenshu), BL54 (Zhibian), SP6 (Sanyinjiao), GB39 (Xuanzhong), and others. Depending on the acupoint, the needle was inserted perpendicularly or obliquely and retained for 15 min. In principle, the acupuncture treatment was performed twice a day during the inpatient stay, but depending on the symptoms of the patient or the circumstances of the patient, the number of acupuncture treatments was adjusted.

Moxibustion and cupping were performed at the acupuncture treatment session. For moxibustion therapy, electronic moxibustion (Technoscience, Seoul, Korea) was administered on the acupoint for 15 min. For cupping therapy, sterile cups (Dongbang Medical, Seongnam, Korea) were applied to 2 points with negative pressure for 15 min. These treatments were performed on meridians related to the pain area.

For herbal medicine therapy, the patients took herbal medicine 30 min after meals twice daily. The herbal medicine was prescribed according to the identification of patterns of the patient’s symptoms by a KMD, and contained blood-circulation-promoting and analgesic agents.

Integrative KM treatment was provided as necessary by the KMD during the hospital stay.

### 2.4. Outcome Measures

#### 2.4.1. Primary Outcome: NRS Score

The NRS is an objective measure of the subjective pain experienced by the patient [33], with 0 points indicating no pain and 10 points indicating severe pain. The NRS scores for LBP were measured three times, once each at the time of admission, discharge, and follow-up, and the measurement was based on the average NRS of the last 3 days.

#### 2.4.2. Secondary Outcomes

##### Oswestry LBP Disability Index (ODI)

The functional status of patients was assessed using the ODI, and the mean score was obtained by dividing the total score by the number of items. The ODI was developed to evaluate lumbar disability during daily life [34]. The ODI includes 10 items, each rated on a scale ranging from 0 to 5 points, for a maximum possible score of 50. Higher ODI scores indicate a more severe disability. This study assessed the ODI at admission, discharge, and follow-up.

##### The 5-Level EuroQol 5-Dimension (EQ-5D-5L)

The EQ-5D-5L was developed to assess health-related QoL (HR-QoL) and is widely used in health care. It consists of five domains (mobility, self-care, usual activities, pain/discomfort, and anxiety/depression) related to current health status, with each domain divided into five levels (level 1: no problem at all, level 2: slight problem, level 3: moderate problem, level 4: severe problem, and level 5: extreme problem). This study calculated the weighted HR-QoL values using the weighted model estimated for Koreans [35], and EQ-5D-5L scores were assessed at admission, discharge, and follow-up.

##### Patient Global Impression of Change (PGIC)

The PGIC is used to assess the subjective impression of change or improvement according to seven levels (1, very much improved; 2, much improved; 3, minimally improved; 4, no change; 5, minimally worse; 6, much worse; and 7, very much worse). The PGIC was initially developed for psychological research; however, it is currently used in various other medical fields to assess changes in pain levels [36]. The PGIC was included in the follow-up survey to evaluate the treatment satisfaction level.

### 2.5. Statistical Analysis

All statistical analyses were two-sided tests, and statistical significance was set at *p* = 0.05. For the basic characteristics of the participants, continuous variables are presented as means and standard deviations, while categorical variables are presented as frequencies and percentages.

The outcomes (NRS, ODI, and EQ-5D scores) at each time point (admission, discharge, and long-term follow-up) are presented as least square means with 95% confidence intervals (CIs) estimated using a linear mixed model. Changes in outcomes from baseline are presented as *p*-values and 95% CIs estimated using a linear mixed model. The analysis was adjusted for baseline outcomes, age, and sex, while time was included as a categorical variable.

Several subgroup analyses, according to age, Cobb angle, and rotational deformity were conducted. Differences in the degree of improvement according to age, Cobb angle, and rotational deformity were determined by analyzing the differences in the reduction in the Cobb angle at different time points using a linear mixed model. The mean baseline values were calculated by pooling data from all patients, while the outcomes at discharge and follow-up for both groups were presented as least squares mean and 95% CI. The differences and *p*-values of the changes in the outcomes between the two groups were calculated using a linear mixed model. The analysis was adjusted for baseline outcomes, age, and sex, while time was included as a categorical variable. In the case of subgroup analysis of participants over the age of 50, only baseline outcomes and age were adjusted. Factors of each subgroup analysis and time were included as cross terms. All changes are expressed as 95% CIs and *p*-values.

For items surveyed at the long-term follow-up, categorical variables are presented as frequencies and percentages, while continuous variables are presented as means and standard deviations.

Factors influencing discharge and long-term follow-up improvement were analyzed using logistic regression models. Improvement was defined as achieving the minimum clinically important difference (MCID). Slight differences exist in the MCID for LBP used in different studies; however, the MCID for the NRS for LBP was set to 1.2. In contrast, the MCID for ODI was set to 12.8, based on a previous study [37] and internal discussion within the research team. A model including basic data (e.g., Cobb angle, sex, age, obesity, drinking status, and smoking) was constructed for multivariate analysis. Each estimate was presented as an odds ratio (OR) and 95% CI, and the area under the curve (AUC) was calculated for each model.

Sensitivity analysis was performed by including outcomes at admission and discharge for all patients, regardless of their participation in the follow-up survey. Among subgroup analysis factors, only Cobb angle was included in sensitivity analysis because we thought Cobb angle was the most representative element. Changes in outcomes from baseline and differences according to Cobb angle were calculated using a linear mixed model for these patients. The factors contributing to improvement at discharge were analyzed using a multivariate logistic regression model. The group that included patients who did not participate in the follow-up survey was the total group. However, the long-term follow-up group had only patients who participated in the follow-up survey.

All analyses were performed using R-4.2.1 for Windows (The R Foundation).

## 3. Results

### 3.1. Participant Selection

Among 1175 patients who visited a KM hospital for TA-induced LBP and were radiologically diagnosed with scoliosis between 1 January 2015, and 30 June 2021, 482 patients were ineligible based on the exclusion criteria, and 19 patients with missing baseline outcome data were excluded. Consequently, the medical records of 674 patients were analyzed. Of these, 407 patients who did not meet the criteria for the follow-up survey were excluded; 267 patients participated in the follow-up survey. The reasons for failing to meet the criteria for the follow-up survey included no permission to receive text messages, foreign nationality, duplicate cell phone numbers, and using a phone number with an area code other than 010. Of these patients, 102 did not respond to the follow-up survey, and 64 refused to participate. Consequently, 101 patients completed the follow-up survey (Figure 2).

### 3.2. Baseline characteristics

The baseline characteristics of the study participants are presented in Table 1. The mean age was 38.69 ± 14.10. Ages <30 and 30~49 each accounted for 35.7%, and age ≥50 accounted for 28.7%. In Cobb angle groups, Cobb angle ≤20° accounted for 88.1% and Cobb angle >20° accounted for 11.9%. In rotational deformity groups, the deformity group accounted for 55.4%, and the non-deformity group accounted for 44.6%. The percentage of women was 54.5%. NRS for LBP at admission was 4.86 ± 0.79 points, EQ-5D-5L score at admission was 0.67 ± 0.15 points, and the ODI score at admission was 35.96 ± 14.82 points. Tendencies were also similar in the total and long-term follow-up groups regardless of Cobb angle (Appendix A).

### 3.3. Treatment Details during the Hospital Stay

Details of the patients’ treatments are shown in Appendix A. The length of stay was similar between the total and long-term follow-up groups (9.24 ± 3.92 and 9.48 ± 4.26 days, respectively). Acupuncture, electropuncture, and cupping therapies were administered to all inpatients, whereas pharmacopuncture, herbal medicine, and Chuna therapies were administered to 98.5%, 99.4%, and 90.9% of patients, respectively. The type and frequency of each treatment were similar between the total and long-term follow-up groups.

### 3.4. Changes in Outcomes from Baseline

Table 2 shows the changes in outcomes at admission, discharge, and follow-up relative to baseline in the long-term follow-up group. The NRS score for LBP decreased from 4.86 (95%CI 4.71 to 5.02) at admission to 3.53 (95%CI 3.17 to 3.90) at discharge and to 3.01 (95%CI 2.64 to 3.38) at follow-up (*p* < 0.001). The ODI score decreased from 35.96 (95%CI 33.08 to 38.85) points at admission to 22.73 (95%CI 20.23 to 25.24) points at discharge and to 14.21 (95%CI 11.74 to 16.67) points at follow-up (*p* < 0.001). EQ-5D-5L scores increased from 0.67 (95%CI 0.64 to 0.70) points at admission to 0.78 (95%CI 0.76 to 0.80) points at discharge and to 0.88 (95%CI 0.86 to 0.90) points at follow-up (*p* < 0.001). Overall, there were significant decreases in NRS scores for LBP and ODI and a significant increase in EQ-5D-5L scores over time. The results were similar for all groups (Appendix A).

### 3.5. Differences in Outcomes According to Age, Cobb Angle, and Ratational Deformity

Table 3 and Figure 3, Figure 4 and Figure 5 show the changes in outcomes according to age, rotational deformity at admission, discharge, and follow-up relative to baseline in the long-term follow-up group.

An analysis of outcomes according to Cobb angle indicated that the change in NRS at discharge was greater in the ≤20° group (diff: −0.13, 95% CI −1.24–0.98), while the change in NRS at follow-up was greater in the >20° group (diff: 0.09, 95%CI −1.03–1.20). For ODI scores, changes in scores at discharge were greater in the >20° group (diff: 0.63, 95%CI −6.53 to 7.80), while changes at follow-up were greater in the ≤20° group (diff: −1.21, 95%CI −8.35 5.94). The differences in the change amount were insignificant between the two groups for any outcomes at any time point (Table 3, Figure 2). Similar tendencies were observed in the total group (Appendix A).

An analysis of outcomes according to rotational deformity indicated that the change in NRS at discharge was greater in the rotation group (diff: −0.03, 95% CI −0.75 to 0.70), while the change in NRS at follow-up was greater in the non-rotation group (diff: 0.15, 95%CI −0.57 to 0.88). For ODI scores, changes in scores at discharge were greater in the rotation group (diff: −1.89, 95%CI −6.61 to 2.83), while changes at follow-up were greater in the non-rotation group (diff: 3.51, 95%CI −1.13 to 8.16). For EQ-5D scores, changes in scores at discharge were greater in the rotation group (diff: 0.02, 95%CI −0.02 to 0.06), while changes at follow-up were greater in the non-rotation group (diff: −0.04, 95%CI −0.08 to 0.00).

An analysis of outcomes according to age indicated that the change in NRS at discharge was greater in the age <50 group (diff: −0.02, 95% CI −0.78 to 0.75), while the change in NRS at follow-up was greater in the age ≥50 group (diff: 2.03, 95%CI 1.27 to 2.80). For ODI scores, changes in scores at discharge and follow-up were greater in the age ≥50 group (diff: 4.60, 95%CI −0.45 to 9.66) (diff: 15.70, 95%CI 10.74 to 20.66). For EQ-5D scores, changes in scores at discharge and follow-up were greater in the age ≥50 group (diff: −0.01, 95%CI −0.06 to 0.03) (diff: −0.09, 95%CI −0.14 to −0.05).

### 3.6. Follow-Up Survey Results

A total of 101 patients were followed-up after discharge. The median time from discharge to long-term follow-up was 749 days (IQR 481, 1205). Two patients were recommended to undergo surgery before hospitalization, of which one patient underwent surgery. Three patients were recommended to use a scoliosis brace, two of which wore it before hospitalization. No patients were recommended to undergo surgery, use a scoliosis brace, or undergo surgery after admission. A total of 87.1% of patients were satisfied with inpatient care, and their preference was slightly higher for KM treatment than for WM treatment. However, eight patients (7.9%) had received treatment for LBP in the preceding 3 months, with most (n = 5) selecting acupuncture therapy. Three patients selected drug therapy involving anti-inflammatory analgesic agents, representing the most common WM treatment. Based on the PGIC, 79.3% of patients reported improved LBP over the next 3 months (Table 4).

### 3.7. Factors Influencing Improvement

Table 5 shows the factors influencing the achievement of the MCID for the NRS and ODI at discharge and follow-up. The number of patients who achieved the MCID in the NRS was 29 at discharge and 68 at follow-up. The number of patients who achieved the MCID in the ODI was 47 at discharge and 78 at follow-up. Patients with a high baseline NRS and BMI ≥25 exhibited higher odds of achieving the MCID in the NRS. In contrast, those ≥30 years and smokers showed lower odds of achieving the MCID. However, there were no significant influencing factors, except for age ≥50 years (OR at follow-up: 0.25, 95% CI: 0.08–0.75). Patients aged 30–49 years exhibited higher odds of achieving the MCID in the ODI, whereas women, those aged ≥50 years, those with a BMI ≥25, drinkers, those with Cobb angle >20°, and those with rotational deformity had lower odds of achieving the MCID in the ODI. However, there were no significant influencing factors, except for age ≥50 years (OR at discharge and follow-up: 0.29, 95% CI: 0.09–0.89; 0.29 95% CI: 0.09–0.91, respectively). The AUC was higher for the NRS model than for the ODI model. In the total group, patients with higher baseline values and BMI ≥25 exhibited higher odds of improving, whereas patients aged 30–49 years exhibited lower odds of recovery than those aged <30 years (Appendix A).

## 4. Discussion

Scoliosis, a condition characterized by misalignment of the vertebral bodies along the coronal plane, is accompanied by rotation of the vertebral bodies in the horizontal plane and loss of curvature in the sagittal plane. Scoliosis is diagnosed based on a curvature ≥10° in the coronal plane in plain spinal radiography [9]. Approximately 60% of patients with scoliosis experience LBP, and LBP tends to be more severe in patients with more severe incurvation [38]. Many studies have examined the effectiveness of KM treatments for scoliosis and LBP; however, none have investigated their efficacy in patients with acute LBP and pre-existing scoliosis. This study assessed the efficacy of integrative KM treatment in patients with TA-induced acute LBP and pre-existing scoliosis.

Analysis of baseline characteristics revealed that most patients were relatively young (mean age, 38 years old), and the highest percentage was in the 20–29 years group. Most patients had mild scoliosis with a Cobb angle of 10–20°, while approximately 12% had a Cobb angle of 21–35°. However, there were no significant differences in the basic characteristics or baseline outcomes between groups having Cobb angles of ≤20° and >20°. Furthermore, there were differences in smoking status and alcohol use, which may have been due to the small sample size of the >20° group (n =12).

Adult scoliosis is divided into adult idiopathic scoliosis and adult degenerative scoliosis. The etiology of adult idiopathic scoliosis is not precisely known, and it is likely that it is a combination of genetic and mechanical factors. Adult degenerative scoliosis usually results from progressive degenerative structural changes in the spine leading to loss of lumbar lordosis in the sagittal plane and development of scoliosis [21]. Ultimately, both of these conditions can lead to high levels of functional disability due to back and leg pain, which leads to a reduced health-related quality of life [39]. Non-surgical treatment currently recommended for these patients is the use of non-steroidal anti-inflammatories and physical therapy [19], but the available evidence on non-surgical treatment in adult scoliosis is extremely limited [40].

Patients with degenerative lumbar scoliosis often have severe back pain, radiating pain in the lower extremities, and symptoms of spinal imbalance. Low back pain is caused by degenerative arthrosis of the facet joint, degenerative changes in the intervertebral disc, and loss of lordosis, and coronal imbalance or sagittal imbalance can also cause back pain. One study reported that people with LBP used more hip and knee joints in the squat position than people without LBP, and that ankle movement was also affected. Thus, low back pain is associated with biomechanical changes in the lower limbs, and these changes may affect low back pain [41]. Lower extremity radiating pain is caused by compression of the nerve root by the pedicle due to spinal stenosis and rotational subluxation [42]. Radiating pain in the lower limb caused by spinal canal stenosis can be treated with decompression surgery, and radiating pain in the lower extremity caused by rotation subluxation can be treated surgically by correcting the rotation subluxation to reduce the pressure applied to the nerve root [43]. However, elderly patients often have medical comorbidities, which affects the incidence of complications after surgery, and the incidence of complications increases with age [44]. Therefore, it is considered important to alleviate symptoms through non-surgical Korean medicine treatment for elderly patients with degenerative lumbar scoliosis.

One case study [29] demonstrated the potential effectiveness of acupuncture treatment on back pain and curvature progression in adult degenerative scoliosis. Another study suggested that Chuna can be an effective treatment method for idiopathic scoliosis as a complementary and alternative therapy [45].

During the inpatient stay, the study participants received integrative KM treatment consisting of acupuncture, herbal medicine, pharmacopuncture, Chuna, cupping, and moxibustion. Acupuncture therapy alleviates pain by inhibiting the activation of pro-inflammatory cytokines and the central nervous system [46]. In the current study, the most commonly prescribed herbal medicine, Anshinjitong-tang, consists of *Rehmanniae Radix*, *AngelicaeGigantis Radix*, and other compounds suggested to promote blood circulation, clear blood stasis, and relieve pain. For pharmacopuncture, herbal extracts were injected into the acupoints. The Shinbaro pharmacopuncture used in this study contained GCSB-5 as the main ingredient, which promotes neuroprotection, neuroregeneration, and recovery of motor function by reducing oxidative stress [47].

The primary and secondary outcomes analyses indicated that patients experienced significant improvements in NRS, ODI, and EQ-5D-5L scores at discharge and follow-up compared with baseline (admission) levels. Of course, in the case of a single cohort study, rather than randomized trials, the interpretation of results should be undertaken with caution because the meaningful magnitude of the change may vary depending on various factors such as the patient population and the length of follow-up [48]. However, even when compared with the results of various studies, for example, that of C Bombardiers et al. [48], the amount of change shown in this study is judged to be significant, particularly that in ODI. These findings indicate that integrative KM treatment can effectively treat patients with TA-induced LBP and pre-existing scoliosis. In particular, the condition at the time of discharge was well retained until the long-term follow-up, with follow-up survey results showing that few patients received additional treatments after discharge. In other words, the patients could maintain pain relief, functional improvements, and QoL improvements in the long term after discharge. Meanwhile, when patients were divided into two groups based on a cutoff Cobb angle of 20°, we observed no differences in the effectiveness of KM treatments between the groups. This may have been due to the small sample size of the >20° group.

Meanwhile, we also performed subgroup analyses according to Cobb angle, age, and rotational deformity. In all subgroups, patients experienced significant improvements in NRS, ODI, and EQ-5D-5L, but there were no differences in the effectiveness of KM treatments between the groups.

An analysis of factors influencing NRS and ODI outcomes indicated that patients ≥50 years had lower odds of achieving the MCID for NRS and ODI, and it was generally more difficult for women to reach the MCID. Various studies have demonstrated that old age is a risk factor for recovery from LBP [49,50]. BMI exerted opposite effects on NRS and ODI, suggesting that patients with overweight status can achieve pain relief relatively easily but have difficulty achieving functional recovery. Patients with Cobb angle >20° and with rotational deformity also had lower odds of achieving recovery; however, the difference was insignificant. Age and BMI were the only factors that significantly influenced the achievement of the MCID, and the effects of these factors on NRS and ODI scores did not show consistent directionality. Based on these findings, it can be inferred that pain and function do not always move in the same direction, necessitating further studies regarding the causative mechanisms for such differences. In addition, the AUC values for the NRS model at discharge and long-term follow-up were 0.68, which indicated a fair explanatory power.

This study’s findings had some limitations. First, the participants were divided into two groups based on Cobb angle; however, the baseline characteristics of the two groups could not be controlled, given the study’s nature. However, because there were no significant differences in the baseline characteristics or outcomes between the two groups, bias related to these factors was considered minimal. Second, most participants in the study were patients with mild scoliosis with an incurvature of 10–20°; therefore, the effectiveness of KM treatment in patients with severe scoliosis could not be identified. Finally, the response rate to Google Forms and telephone surveys was low due to changes in contact numbers, even after three or more attempts to reach the participants by telephone. The sample size of the final analysis was relatively small. This sample size was further reduced by separating the sample into subgroups, so care should be taken when interpreting the results. Furthermore, because it was an exploratory study and we included patients who visited the hospital for a specific period of time, there was a lack of logical basis for the sample size of 101. However, we observed no significant differences in baseline characteristics, treatment details, or treatment outcomes between the total and follow-up groups. This suggests that the patients who participated in the follow-up survey represented the entire study population.

Notably, the present study is the first follow-up study of patients with scoliosis who received integrative KM treatment for acute LBP. Our findings demonstrated that integrative KM treatment effectively reduces LBP, improves functional impairment, and enhances QoL in patients with acute LBP and pre-existing mild scoliosis. In addition, KM treatment showed the same effect regardless of age, Cobb angle, and rotational deformity. This means that KM treatment can be an effective treatment for various forms of mild scoliosis in adults. Moreover, the follow-up survey results confirmed that patients were highly satisfied with KM treatment, and the pattern of improvements achieved through integrative KM treatment was retained until the follow-up survey. Future large-scale clinical trials that include patients with varying severities of scoliosis are required.

## 5. Conclusions

This study found that integrative KM treatment can reduce LBP, improve lumbar dysfunction, and enhance the QoL in patients with TA-induced acute LBP and pre-existing mild scoliosis. More research is needed to confirm this finding.

## Figures and Tables

**Figure 1 healthcare-11-01735-f001:**
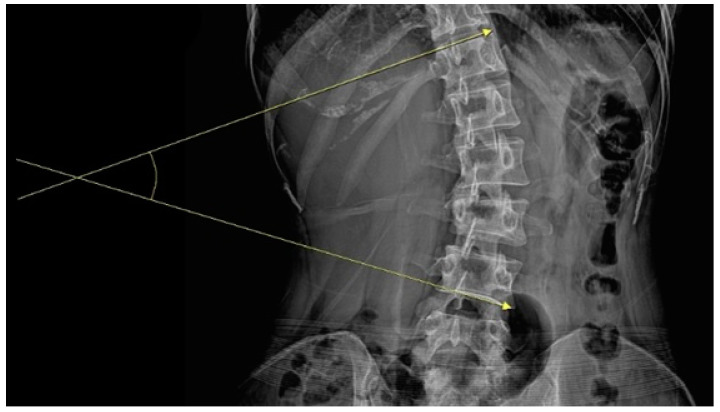
Measurement of Cobb angle.

**Figure 2 healthcare-11-01735-f002:**
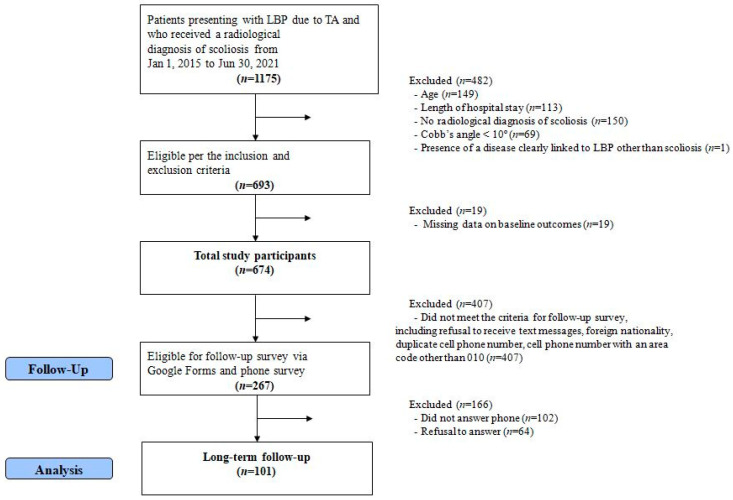
Flow chart of patient selection.

**Figure 3 healthcare-11-01735-f003:**
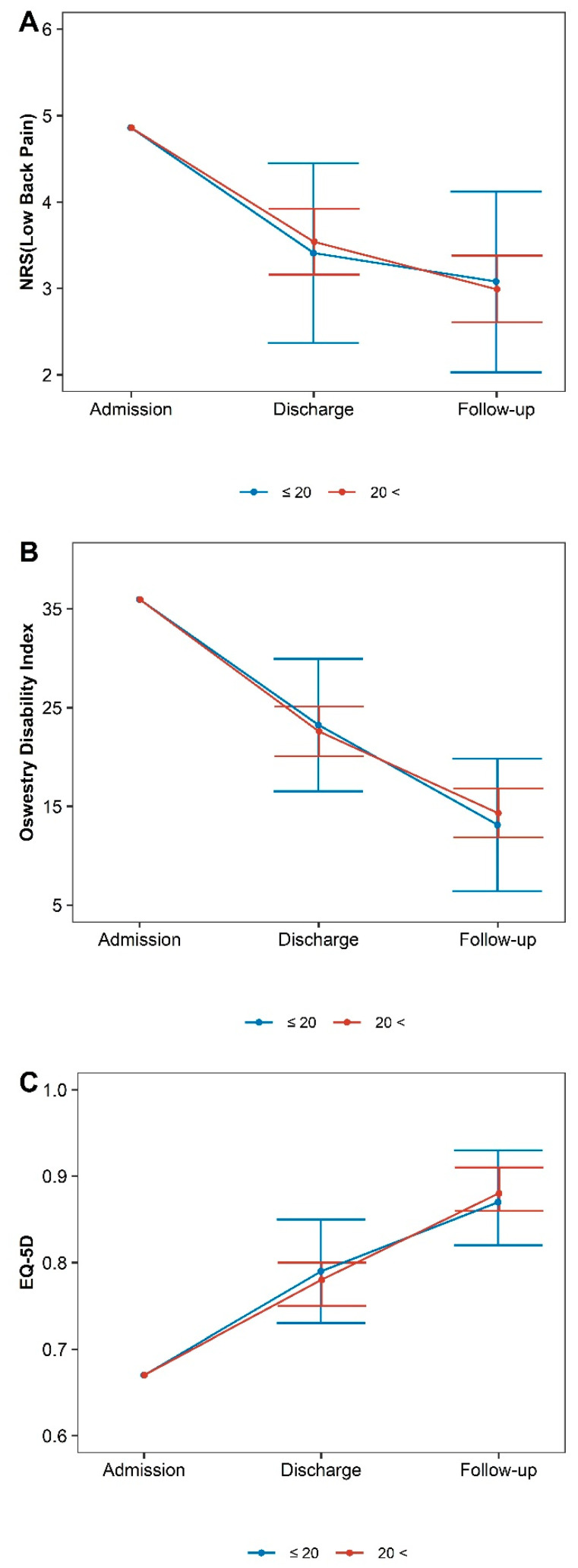
Differences in outcomes according to Cobb angle. (**A**) Numeric rating scales for low back pain, (**B**) Oswestry Disability Index, (**C**) European Quality of Life 5 Dimensions 5 Level Version.

**Figure 4 healthcare-11-01735-f004:**
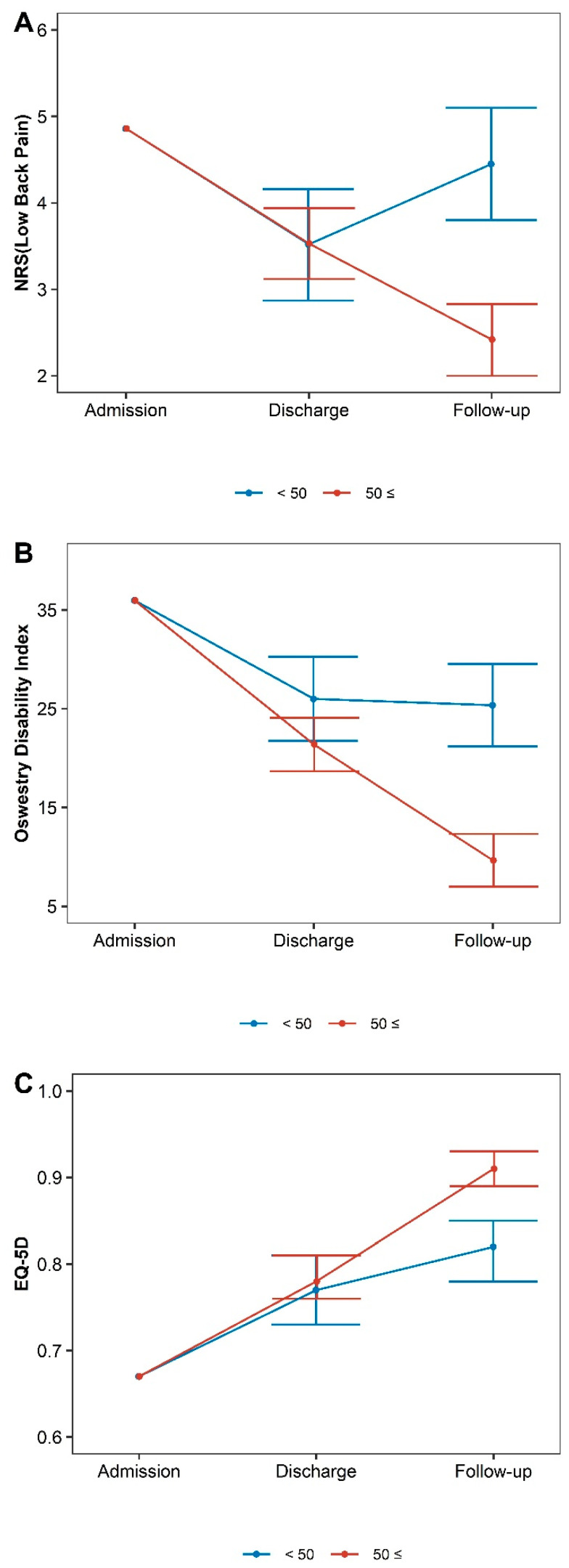
Differences in outcomes according to age. (**A**) Numeric rating scales for low back pain, (**B**) Oswestry Disability Index, (**C**) European Quality of Life 5 Dimensions 5 Level Version.

**Figure 5 healthcare-11-01735-f005:**
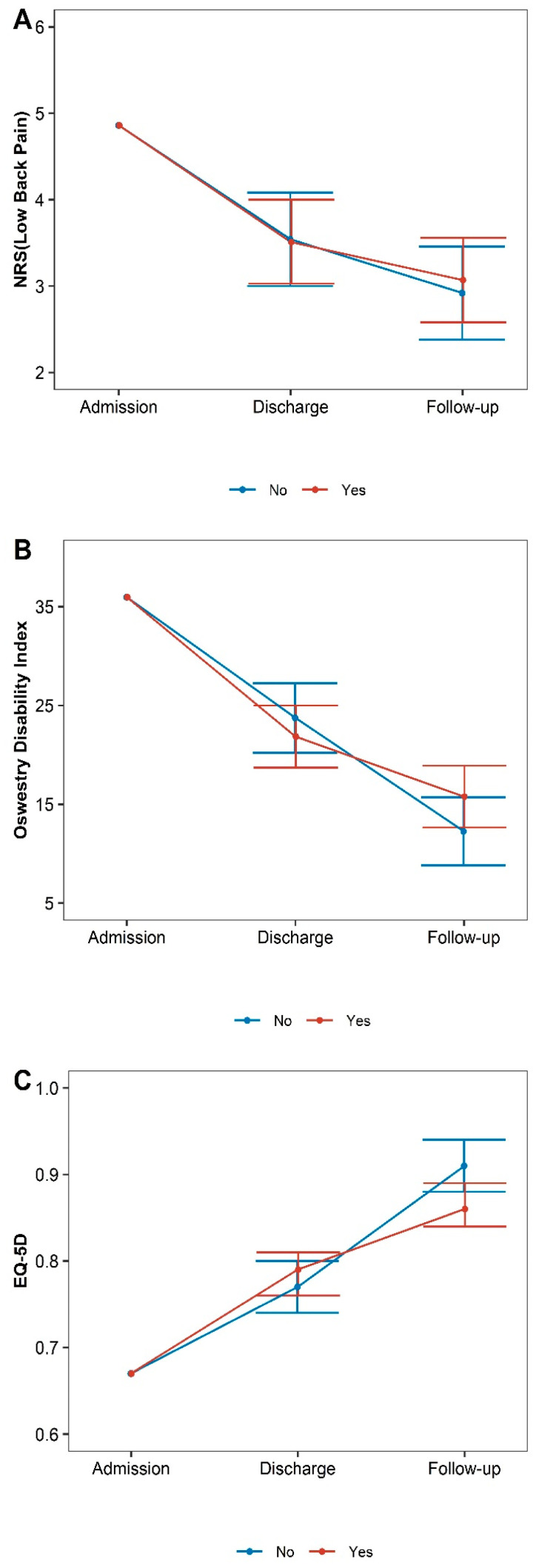
Differences in outcomes according to rotational deformity. (**A**) Numeric rating scales for low back pain, (**B**) Oswestry Disability Index, (**C**) European Quality of Life 5 Dimensions 5 Level Version.

**Table 1 healthcare-11-01735-t001:** Basic characteristics of study participants.

	Total
	N = 101
Age	
Mean ± SD	38.69 ± 14.10
<30	36 (35.7)
30–49	36 (35.7)
≥50	29 (28.7)
Cobb’s Angle	
≤20	89 (88.1)
>20	12 (11.9)
Rotational deformity	
Yes	56 (55.4)
No	45 (44.6)
Sex	
Male	46 (45.5)
Female	55 (54.5)
Height (Total N = 670)	
Mean ± SD	166.29 ± 9.05
Weight (Total N = 670)	
Mean ± SD	66.21 ± 15.70
BMI (Total N = 670)	
≤25	61 (60.4)
>25	40 (39.6)
Comorbidity	
Hypertension	12 (11.9)
Diabetes mellitus	3 (3.0)
Depressive disorder	4 (4.0)
Cardiovascular disease	10 (9.9)
Respiratory disease	6 (5.9)
Gastrointestinal disease	19 (18.8)
Others	42 (41.6)
Smoking	
Yes	24 (23.8)
No	77 (76.2)
Alcohol use	
Yes	36 (35.6)
No	65 (64.4)
Outcomes	
LBP for NRS	4.86 ± 0.79
EQ-5D-5L	0.67 ± 0.15
ODI	35.96 ± 14.82
ROM—Flexion (Total N = 671)	89.01 ± 6.56
ROM—Extension (Total N = 671)	19.60 ± 2.42

Abbreviations: BMI: body mass index; LBP: low back pain; EQ-5D-5L: 5-level EuroQol 5-dimension: ODI: Oswestry Disability Index; ROM: range of motion.

**Table 2 healthcare-11-01735-t002:** Changes in outcomes relative to baseline in the long-term follow-up groups.

	Admission (Baseline)	Discharge	Follow-Up
NRS for LBP			
Value	4.86 (4.71 to 5.02)	3.53 (3.17 to 3.90)	3.01 (2.64 to 3.38)
Diff	—	1.33 (0.96 to 1.69)	1.85 (1.48 to 2.22)
*p* value		<0.001	<0.001
Cohen’s d		2.15	0.71
ODI			
Value	35.96 (33.08 to 38.85)	22.73 (20.23 to 25.24)	14.21 (11.74 to 16.67)
Diff	—	13.23 (10.73 to 15.73)	21.76 (19.29 to 24.23)
*p* value		<0.001	<0.001
Cohen’s d		0.96	1.14
EQ-5D-5L			
Value	0.67 (0.64 to 0.70)	0.78 (0.76 to 0.80)	0.88 (0.86 to 0.90)
Diff	—	−0.11 (−0.13 to −0.09)	−0.21 (−0.23 to −0.19)
*p* value		<0.001	<0.001
Cohen’s d		−0.84	−1.12

Values are presented as least square means and 95% confidence intervals. Differences from baseline and *p*-values were calculated using a linear mixed model adjusted for baseline outcome, sex, and age. Abbreviations: NRS, numeric rating scale; ODI, Oswestry Disability Index; EQ-5D-5L, 5-level EuroQol 5-dimension; Diff, the difference from baseline.

**Table 3 healthcare-11-01735-t003:** Differences in outcomes according to age, Cobb angle, and rotational deformity.

		Admission (Baseline)	Discharge	Follow-Up
NRS_LBP	Age <50	4.86 (4.71 to 5.02)	3.52 (2.87 to 4.16)	4.45 (3.80 to 5.10)
	Age ≥50		3.53 (3.12 to 3.94)	2.42 (2.00 to 2.83)
	Difference	—	−0.02 (−0.78 to 0.75)	2.03 (1.27 to 2.80)
	*p* value	—	0.969	<0.001
	Cobb angle ≤ 20°	4.86 (4.71 to 5.02)	3.41 (2.37 to 4.45)	3.08 (2.03 to 4.12)
	Cobb angle > 20°		3.54 (3.16 to 3.92)	2.99 (2.61 to 3.38)
	Difference	—	−0.13 (−1.24 to 0.98)	0.09 (−1.03 to 1.20)
	*p* value	—	0.816	0.88
	rotation X	4.86 (4.71 to 5.02)	3.54 (3.00 to 4.08)	2.92 (2.38 to 3.46)
	rotation O		3.51 (3.03 to 4.00)	3.07 (2.58 to 3.56)
	Difference	—	−0.03 (−0.75 to 0.70)	0.15 (−0.57 to 0.88)
	*p* value	—	0.943	0.68
ODI	Age <50	35.96 (33.08 to 38.85)	25.99 (21.73 to 30.26)	25.35 (21.17 to 29.54)
	Age ≥50		21.39 (18.68 to 24.10)	9.65 (6.98 to 12.33)
	Difference	—	4.60 (−0.45 to 9.66)	15.70 (10.74 to 20.66)
	*p* value	—	0.074	<0.001
	Cobb angle ≤20°	35.96 (33.08 to 38.85)	23.23 (16.53 to 29.94)	13.12 (6.42 to 19.83)
	Cobb angle >20°		22.60 (20.08 to 25.12)	14.33 (11.85 to 16.81)
	Difference	—	0.63 (−6.53 to 7.80)	−1.21 (−8.35 to 5.94)
	*p* value	—	0.861	0.74
	rotation X	35.96 (33.08 to 38.85)	23.74 (20.22 to 27.26)	12.26 (8.82 to 15.69)
	rotation O		21.85 (18.70 to 24.99)	15.77 (12.65 to 18.89)
	Difference	—	−1.89 (−6.61 to 2.83)	3.51 (−1.13 to 8.16)
	*p* value	—	0.431	0.137
EQ5D	Age <50	0.67 (0.64 to 0.70)	0.77 (0.73 to 0.81)	0.82 (0.78 to 0.85)
	Age ≥50		0.78 (0.76 to 0.81)	0.91 (0.89 to 0.93)
	Difference	—	−0.01 (−0.06 to 0.03)	−0.09 (−0.14 to −0.05)
	*p* value	—	0.575	<0.001
	Cobb angle ≤20°	0.67 (0.64 to 0.70)	0.79 (0.73 to 0.85)	0.87 (0.82 to 0.93)
	Cobb angle >20°		0.78 (0.75 to 0.80)	0.88 (0.86 to 0.91)
	Difference	—	0.02 (−0.05 to 0.08)	−0.01 (−0.07 to 0.05)
	*p* value	—	0.605	0.773
	rotation X	0.67 (0.64 to 0.70)	0.77 (0.74 to 0.80)	0.91 (0.88 to 0.94)
	rotation O		0.79 (0.76 to 0.81)	0.86 (0.84 to 0.89)
	Difference	—	0.02 (−0.02 to 0.06)	−0.04 (−0.08 to 0.00)
	*p* value	—	0.411	0.041

The mean baseline outcomes were calculated by pooling the baseline values from both groups. The outcomes of the two groups at discharge and follow-up are presented as least square means and 95% CIs estimated using a linear mixed model. Difference refers to the change from the baseline for the two groups. Differences and *p*-values were calculated using a linear mixed model adjusted for baseline outcomes, sex, and age. Abbreviations: NRS, numeric rating scale; ODI, Oswestry Disability Index; EQ-5D, 5-level EuroQol 5-dimension.

**Table 4 healthcare-11-01735-t004:** Results of the long-term follow-up survey.

	n = 101
Recommendation for surgery before hospitalization	
Yes	2 (2.0)
No	99 (98.0)
Low back surgery before hospitalization	
Yes	1 (1.0)
No	100 (99.0)
Type of low back surgery before hospitalization	
Anterior/posterior spinal fusion	-
Anterior/posterior epiphysiodesis	-
Hemivertebral resection	-
Others	-
Surgery type unknown	1 (100.0)
Recommendation to use a scoliosis brace before hospitalization	3 (3.0)
Use of a scoliosis brace before hospitalization	2 (2.0)
Recommendation to undergo low back surgery after inpatient treatment	0 (0.0)
Low back surgery after inpatient treatment	0 (0.0)
Recommendation to use a scoliosis brace after inpatient treatment	0 (0.0)
Use of a scoliosis brace after inpatient treatment	0 (0.0)
Satisfaction with inpatient treatment	
Very satisfied	49 (48.5)
Satisfied	39 (38.6)
Neutral	11 (10.9)
Unsatisfied	1 (1.0)
Very Unsatisfied	1 (1.0)
Preference for KM treatment	
Mean ± SD	8.08 ± 1.91
Preference for WM treatment	
Mean ± SD	7.66 ± 1.81
Treatment in the past 3 months	
Yes	8 (7.9)
No	93 (92.1)
Types of treatment in the past 3 months	
Acupuncture	5 (62.5)
Pharmacopuncture	3 (37.5)
Herbal medicine	**-**
Chuna	1 (12.5)
Cupping	2 (25.0)
Moxibustion	3 (37.5)
Cervical retraction	-
Herbal wrapping	-
Injection	-
Drug therapy	3 (37.5)
Exercise therapy	-
Manual therapy	1 (12.5)
Physical therapy	3 (37.5)
Brace	-
Others	-
Don’t know	-
PGIC	
Very much improved	25 (24.8)
Much improved	22 (21.8)
Minimally improved	33 (32.7)
No change	16 (15.8)
Minimally worse	3 (3.0)
Much worse	1 (1.0)
Very much worse	1 (1.0)

Abbreviations: PGIC, patients’ global impression of change; KM, Korean medicine; WM, Western medicine; SD, standard deviation.

**Table 5 healthcare-11-01735-t005:** Factors associated with improvement in the NRS score for LBP and ODI.

	NRS		ODI	
	Discharge (n =101)	Follow-Up (n =101)	Discharge (n =101)	Follow-Up (n =101)
Patients achieving the MCID	29	68	47	78
Baseline value	1.76 (0.89–3.51)	1.31 (0.72–2.37)	1.58 (0.88–2.85)	0.93 (0.49–1.80)
Age (ref. <30 years)				
30–49 years	0.45 (0.14–1.45)	0.47 (0.16–1.43)	2.01 (0.67–6.050)	3.03 (0.66–13.82)
≥50 years	0.97 (0.33–2.85)	0.25 (0.08–0.75)	0.29 (0.10–0.89)	0.30 (0.09–0.96)
Sex (ref. male)				
Female	0.52 (0.18–1.46)	1.06 (0.38–2.92)	0.51 (0.18–1.48)	0.41 (0.12–1.40)
BMI (ref. BMI < 25)				
BMI ≥ 25	1.22 (0.45–3.30)	2.03 (0.76–5.42)	0.29 (0.10–0.83)	0.74 (0.24–2.33)
Smoking (ref. No)				
Yes	0.48 (0.14–1.63)	0.58 (0.18–1.83)	0.59 (0.19–1.80)	1.56 (0.37–6.57)
Drinking (ref. No)				
Yes	0.91 (0.31–2.65)	1.41 (0.50–4.01)	0.29 (0.10–0.81)	0.27 (0.08–0.93)
Cobb angle (ref. <20°)				
>20°	1.97 (0.49–7.99)	0.93 (0.22–3.87)	0.29 (0.06–1.33)	0.94 (0.18–4.88)
rotational deformity (ref. No)				
Yes	0.95 (0.34–2.64)	1.00 (0.39–2.57)	0.86 (0.33–2.26)	0.46 (0.14–1.45)
AUC (95% CI)	0.68 (0.56–0.79)	0.68 (0.56–0.79)	0.53 (0.40–0.67)	0.52 (0.39–0.65)

MCID: minimum clinically important difference; BMI: body mass index; AUC: area under the curve.

## Data Availability

The data presented in this study are available on request from the corresponding author. The data are not publicly available due to privacy/ethical restrictions.

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
