# Peer review of "Effectiveness of Integrative Korean Medicine Treatment in Patients with Traffic-Accident-Induced Acute Low Back Pain and Mild Adult Scoliosis"

_healthcare, 2023, doi:10.3390/healthcare11121735_

Round 1

Reviewer 1 Report (Previous Reviewer 2)

Thank you for resubmitting this manuscript for my review, I see that you have done a lot of work to improve this manuscript. 

There is one aspect that is of particular concern to me and that is that the improvements you have objected to are not clinically relevant to pain. I attach reference:

Bombardier, C. L. A. I. R. E., Hayden, J. I. L. L., & Beaton, D. E. (2001). Minimal clinically important difference. Low back pain: outcome measures. The Journal of Rheumatology28(2), 431-438.

Please include this reference and discuss them. This should also lead to a change in your conclusions.

Best regards

Author Response

[reviewer1]

Dear reviewer1,

We did our best to reflect your opinions as much as possible and to raise our paper quality to suitable for publishing at Healthcare. And we are very delighted an grateful for your kind comments. We appreciate again your further comments and have striven to incorporate them in our revised manuscript again. In the following sections, please find our responses to each of your comments and suggestions.

  1. Thank you for resubmitting this manuscript for my review, I see that you have done a lot of work to improve this manuscript.
    There is one aspect that is of particular concern to me and that is that the improvements you have objected to are not clinically relevant to pain. I attach reference:
    Bombardier, C. L. A. I. R. E., Hayden, J. I. L. L., & Beaton, D. E. (2001). Minimal clinically important difference. Low back pain: outcome measures. The Journal of Rheumatology, 28(2), 431-438.
    Please include this reference and discuss them. This should also lead to a change in your conclusions.
    Best regards
  • Response: Thank you for your kind comments and thanks for providing the reference. We carefully reviewed the references you sent and added to the consideration that our ODI improvement was meaningful, but caution is needed in interpretation. And in this study, there was no significant difference in ODI improvement according to age, cobb's angle, or displacement. We added the above in detail in Discussion part. (21p, 1st and 2nd paragraph)

The primary and secondary outcomes analysis indicated that patients experienced significant improvements in NRS, ODI, and EQ-5D-5L scores at discharge and follow-up compared with baseline (admission) levels. Of course, in case of single cohort study, not randomized trials, the interpretation of results should be cautious, because the meaningful magnitude of the change may vary depending on svarious factors such as the patient population, the length of follow-up, etc [46]. However, even when compared with the results of various studies shown as examples in the C Bombardiers et al. [46], the amount of change shown in this study is judged to be a significant value, particularly in ODI. These findings indicate that integrative KM treatment can effectively treat patients with TA-induced LBP and pre-existing scoliosis. In particular, the condition at the time of discharge was well retained until the long-term follow-up, with follow-up survey results showing that few patients received additional treatments after discharge. In other words, the patients could maintain pain relief, functional improvements, and QoL improvements in the long term after discharge. Meanwhile, when patients were divided into two groups based on a cutoff Cobb’s angle of 20º, we observed no differences in the effectiveness of KM treatments between the groups. It may have been due to the small sample size of the > 20º group.

Meanwhile, we also performed a subgroup analysis according to Cobb’s angle, age, and rotational deformity. In all subgroups, patients experienced significant improvements in NRS, ODI, and EQ-5D-5L, but there were no differences in the effectiveness of KM treaments between the groups. (21p, 1st and 2nd paragraph).

Reviewer 2 Report (New Reviewer)

Congratulations on the idea. The work in my opinion is interesting.  I have the following comments, which I ask you to apply - it will improve the interest in the paper. I have no major substantive comments.

‘’ Adult scoliosis is divided into adult idiopathic scoliosis and adult (…)’’  - I suggest adding general epidemiological data on scoliosis. DOI: 10.1007/s00586-020-06453-0

‘’ Adult scoliosis is divided into adult idiopathic scoliosis and adult degenerative sco-
liosis[13]. (…) Degenerative scoliosis is (…)’’ - I also suggest combining these three paragraphs into one paragraph.

2.5. Statistical analysis - Please complete the information in this paragraph with: what test was used to check the distribution. I would ask you to add the sample size calculation - if there was none for the information in the limitation paragraph. To the ''p'' value, I suggest adding an effect size calculation

4. Discussion - ‘’Patients with degenerative lumbar scoliosis often have severe back pain (….)Korean medicine treatment for elderly patients with degenerative lumbar scoliosis. ‘’  - It should be noted in this paragraph that back pain is associated with a change in biomechanics - the lower extremities. Different work of the lower extremities will exacerbate the pain and pathology of the spine. This will create a reciprocal interaction – I suggest a reference to the paper DOI: 10.26444/aaem/117708

Dear Editor, I see that the paper must have already passed some review. In my opinion, it is well written. I suggested minor developments.  Greetings

Author Response

[Reviewer2]

Dear reviewer2,

We appreciate your kind words and detailed comments and have striven to incorporate them in our revised version of the manuscript. Your suggestions were extremely helpful in improving our paper. In the following sections, please find our responses to each of your comments and suggestions.

  1. ‘’ Adult scoliosis is divided into adult idiopathic scoliosis and adult (…)’’ - I suggest adding general epidemiological data on scoliosis. DOI: 10.1007/s00586-020-06453-0 ‘’ Adult scoliosis is divided into adult idiopathic scoliosis and adult degenerative sco-liosis[13]. (…) Degenerative scoliosis is (…)’’ - I also suggest combining these three paragraphs into one paragraph.
  • Response: Thank you for your kind comments. We added a description of the general epidemiological data on scoliosis. Also, as advised, we have combined the three paragraphs into one paragraph. Thanks for the good advice. (2p, 3rd paragraph)

Adult scoliosis is divided into adult idiopathic scoliosis and adult degenerative sco-liosis[13]. However, it is difficult to clearly distinguish between idiopathic scoliosis and degenerative scoliosis if the patient is not sure when the transformation began[14]. Ac-cording to one study, the prevalence of primary degenerative scoliosis was approximately 38%, more common in women than men (41.2% vs 27.5%), and with a higher prevalence in those over 60 years of age (36% vs 13%). [1] Idiopathic scoliosis is classified according to the patient's age at the time of diagnosis, and can be subdivided into three types. Infantile scoliosis is divided into under 3 years old, juvenile scoliosis is 3-9 years old, and adolescent scoliosis is 10-18 years old [13]. Degenerative scoliosis is a scoliosis developed after adulthood, and the age standard is ambiguous. One study of degenerative scoliosis was based on a minimum age of 45 [15], and another study was based on a minimum age of 50[16]. (2p, 3rd paragraph).

  1. 2.5. Statistical analysis - Please complete the information in this paragraph with: what test was used to check the distribution. I would ask you to add the sample size calculation - if there was none for the information in the limitation paragraph. To the ''p'' value, I suggest adding an effect size calculation
  • Response: Thank you for your sharp points. We rechecked the statistical analysis part according to your comments. First, our sample size was 101, and it was assumed that the central limit theorem was satisfied. Therefore, we conducted no separate test to confirm the distribution. We ask for your understanding on this point. Second, this study is an exploratory and single arm study, including patients who visited the hospital for specific period of time. Therefore, we judged that sample size calculation is not needed. Nevertheless, we think that the lack of logical basis for 101 people is a limitation of this study, and this was added to the limitations of the study. (21p, the last paragraph) Finally, we have added the Cohen’s d values in table2 to as your comments. (Table2)

This study’s findings had some limitations. First, the participants were divided into two groups based on Cobb’s angle; however, the baseline characteristics of the two groups could not be controlled, given the study’s nature. However, because there were no significant differences in the baseline characteristics or outcomes between the two groups, bias related to these factors was considered minimal. Second, most participants in the study were patients with mild scoliosis with an incurvature of 10–20º; therefore, the effectiveness of KM treatment in patients with severe scoliosis could not be identified. Finally, the response rate to Google Forms and telephone surveys was low due to changes in contact numbers, even after three or more attempts to reach the participants by telephone. The final analysis’s sample size was relatively small. This sample size has been further reduced by separating subgroup, so we need to be careful about the interpretation of the results. Also, because it is an exploratory study and we included patients who visited the hospital for specific period of time, there was a lack of logical basis for 101 sample size. However, we observed no significant differences in baseline characteristics, treatment details, or treatment outcomes between the total and follow-up groups. This suggests that the patients who participated in the follow-up survey represented the entire study population. (21p, the last paragraph)

  1. 4. Discussion - ‘’Patients with degenerative lumbar scoliosis often have severe back pain (….)Korean medicine treatment for elderly patients with degenerative lumbar scoliosis. ‘’ - It should be noted in this paragraph that back pain is associated with a change in biomechanics - the lower extremities. Different work of the lower extremities will exacerbate the pain and pathology of the spine. This will create a reciprocal interaction – I suggest a reference to the paper DOI: 10.26444/aaem/117708
  • Response: Thank you for your valuable opinion. We added a description of the reciprocal interaction of low back pain and lower extremity movements. Thanks for the good advice. (20p, 3rd paragraph)

Patients with degenerative lumbar scoliosis often have severe back pain, radiating pain in the lower extremities, and symptoms of spinal imbalance. Low back pain is caused by degenerative arthrosis of the facet joint, degenerative changes in the interver-tebral disc, and loss of lordosis, and coronal imbalance or sagittal imbalance can also cause back pain. One study reported that people with LBP used more hip and knee joints in the squat position than people without LBP, and that ankle movement was also affected. Thus, low back pain is associated with biomechanical changes in the lower limbs, and these changes may affect low back pain[2]. Lower extremity radiating pain is caused by compression of the nerve root by the pedicle due to spinal stenosis and rotational subluxation[40]. Radiating pain in the lower limb caused by spinal canal stenosis can be treated with decompression surgery, and radiating pain in the lower extremity caused by rotation subluxation can be treated surgically by correcting the rotation subluxation to reduce the pressure applied to the nerve root[41]. However, elderly patients often have medical comorbidities, which affects the incidence of complications after surgery, and the incidence of complications increases with age [42]. Therefore, it is considered important to alleviate symptoms through non-surgical Korean medicine treatment for elderly patients with degenerative lumbar scoliosis.  (20p, 3rd paragraph)

  1. McAviney, J., et al., The prevalence of adult de novo scoliosis: A systematic review and meta-analysis. European Spine Journal, 2020. 29: p. 2960-2969.
  2. Zawadka, M., et al., Altered squat movement pattern in patients with chronic low back pain. Annals of Agricultural and Environmental Medicine, 2021. 28(1): p. 158.

Round 2

Reviewer 1 Report (Previous Reviewer 2)

Thank you very much for your corrections for improvement, methodologically it has improved.

Regards

This manuscript is a resubmission of an earlier submission. The following is a list of the peer review reports and author responses from that submission.

Round 1

Reviewer 1 Report

The article is very well written, but I have a major concern about the study sample. The inclusion criteria in terms of age (19-65 years old) and of radiographic diagnosis of scoliosis (>10° cobb) are too much aspecific.

For the first point, a scoliosis in a 20 yo patient is generally idiopathic, while a scoliosis in a 60 yo patient can be idiopathic or degenerative. These are two completely different conditions that cannot be evaluated as a whole in a study. They are simply two different diseases.

For the second point, the cutoff of 10° is too aspecific and the distinction of the results with a 20° cutoff doesn’t add much. In fact, as correctly stated, most participants in the study were patients with mild scoliosis with an incurvature of 10–20º. When we consider that these patients had low back pain related to traffic accidents, it is clear that most of the patients in the included study are probably affected by an antalgic scoliosis, which is not a true scoliotic deformity. Moreover, the authors stated that the patients were affected by a pre-existing scoliosis, but it is not reported how they knew this information. For example, did they have a pre-existing xray, which showed a 12- or 15°-degrees scoliosis for each patient? Or, more probably, did they only have a xray taken after the accident that showed a >10° scoliosis? In that case, I think that most of the included patients are not truly affected by scoliosis (preexisting curve >10°), but, maybe, they had a spinal paramorphism (pre existing scoliotic curve<10°) which is very common (but is not a true scoliotic deformity), and they subsequently had an accident that induced an asymmetric contracture of the back muscles which induced an antalgic scoliosis >10°. In this view, I found that all the background of the article and the conclusions of the article are misleading.

Reviewer 2 Report

First of all, congratulations for the research work done, I am aware of how difficult it is to conduct research without financial support. I will now mention just one recommendation for clearer and more precise information on your results. Regarding the methodology, I have to congratulate you on your work.

In the summary section I recommend not to use abbreviations like KM or TA and to use the full term.

In the treatment section I recommend a more detailed description of the treatment, as well as indicating the order of application and the days of duration of each treatment. I recommend the use of a treatment protocol table. The detail of the treatment during the hospital stay is not sufficient and does not establish a clear treatment protocol to be evaluated. Such a well-defined protocol would avoid reproducing or replicating the article, which would be a serious mistake for publication.

In the primary outcome section, it is recommended to indicate whether the pain measurement is done only once in each period or averaged over several days.

The discussion is more about restating the results than discussing or comparing them with other studies. Proof of this is that less than 5 references are used throughout the discussion. I advocate a reorganisation of the discussion and increase the depth of the comparisons of your results with previous studies.

Due to the large loss of samples during the study, the conclusion should be more cautious, for example, this study found that integrative KM treatment CAN REDUCE low back pain, improve low back dysfunction and improve quality of life in patients with existing acute low back pain - mild scoliosis. MORE RESEARCH IS NEEDED TO CONFIRM THIS SUSPICION.